# Impact of economic recessions on healthcare workers and their crises' responses: study protocol for a systematic review of the qualitative and quantitative evidence for the development of an evidence-based conceptual framework

Tiago Silva Jesus [1], Elias Kondilis [2], Jonathan Filippon [2], Giuliano Russo[2]

For numbered affiliations see end of article.

**Correspondence to**
Dr Tiago Silva Jesus;
jesus-ts@outlook.com

## ABSTRACT

**Introduction** During economic recessions, health professionals face reduced income and labour opportunities, hard conditions often exacerbated by governments' policy responses to crises. Growing evidence points to non-negligible effects on national health workforces and health systems—decrease in motivation, burnout, migration—arising from the combination of crisis-related factors. However, no theoretical conceptualisation currently exists framing the impacts recessions have on human resources for health (HRH), or on their reactions.

**Methods and analysis** This paper lays out a protocol for a systematic review of the existing qualitative, quantitative and mixed-method evidence on the economic recessions and HRH; results from the review will be used to develop a conceptual framework linking existing theories on recessions, austerity measures, health systems and population health, with a view of informing future health policies. Eight relevant databases within the health, health systems, multidisciplinary and economic literature will be searched, complemented by secondary searches and experts' input. Eligible studies will present primary quantitative or qualitative evidence on HRH impacts, or original secondary analyses. We will cover the 1970–2019 period—the modern age of global economic recessions—and full texts in English, Spanish, Italian, French, Portuguese or Greek. Two reviewers will independently assess, perform data extraction and conduct quality appraisal of the texts identified. A 'best-fit' framework synthesis will be applied to summarise the findings, using an a priori, theoretically driven framework. That preliminary framework was built by the research team to inform the searches, and will be appraised by external experts.

**Ethics and dissemination** In addition to peer-reviewed publications, the new framework will be presented in global health systems research conferences and inform regional policy dialogue workshops in Latin America on economic recessions and health systems.

---

**Strengths and limitations of this study**

► Currently debated topic, drawing from the growing literature on economic recessions and health.
► Inclusion of databases for the mainstream health, health systems and economic literature and inclusion of studies with full texts in six languages, to pointedly cover countries recently hit by economic recessions.
► Use of an a priori, theoretically grounded framework revised by experts to guide the data collection, and leading to an evidence-based framework on economic recessions and human resources for health.
► We do not review unpublished evidence or articles without an abstract in English.
► Apart from consulting experts, we do not use other forms of public involvement in this systematic review.

---

**PROSPERO registration number** CRD42019134165.

## INTRODUCTION

Economic recessions have been a recurrent phenomenon during the last decades in high-income as well as low-income and middle-income countries (LMICs). Following the 2008 financial crisis in the USA, world economies have experienced a period of economic instability (the so-called 'Great Recession') and deteriorating health outcomes, spreading first to Europe,[1] Africa[2] and more recently to South America.[3] There is enough evidence suggesting that economic contractions do affect the health and healthcare of populations,[3–7] although this effect is not always straightforward and seems to depend on the policy responses to the crises.[4–6 8]

**BMJ**

As other workers during an economic recession, healthcare workers are affected by a combination of job insecurity, decreased purchasing power and reduced labour market opportunities.[9] It has also been suggested that these effects are often amplified by the reduction of welfare support and salary cuts by the restrictive policies governments typically apply to the largest spending sectors (including commonly health) in the attempt to balance budgets and reduce deficits.[8 10]

The known effects of economic contractions and austerity policies on health workers include salary cuts and job losses, compounding migration intentions,[11 12] decreased motivation,[10] unwanted organisational changes[13] and an increase in the tendency to engage in concurring profit-generating activities,[14] often at the expense of the quality of service. Scholars have started to explore the area of the effect of Europe's most recent financial crisis on human resources for health (HRH) policies,[15] and health workers' responses to the changing economic circumstances,[10 11 16] but to the best of our knowledge, no specific review has been conducted on this subject.

The aim of the study is to review the evidence on the impacts of economic recessions on HRH, and use it to draw a framework for the conceptualisation of the repercussions. We aim to do so through a synthesis of both the qualitative and quantitative evidence of the impact of economic recessions on the HRH. In particular, we are interested in reviewing qualitative and quantitative evidence on: (1) any direct or indirect impacts of economic recessions on the HRH (salary cuts, loss of motivation, reorganisation of provision of services, etc), including identifying any mediating or moderating variables, and the (2) health workers' reactions to such effects (migration, engagement in alternative profit generating activities, early retirement, etc). The evidence arising from each subject will be combined into a single, yet comprehensive, evidence-based conceptual framework of the conjunct of effects of economic recessions on the HRH.

In that context, our specific study questions are
A. What have been the key impacts of economic recessions, occurred since 1970, for the global and national health workforces, and through which mechanisms, mediating or moderating variables have those impacts occurred?
B. How have HRH in different contexts reacted (ie, emigrated, became involved into dual practice, etc) to the consequences of economic recessions, or to any resultant policy measures, such as but not limited to the so-called austerity measures?

It is hoped that the framework from the uncovered evidence will strengthen the current understanding of this complex socio, political, economic and labour market phenomenon towards informing both ex ante and ex post policies on preventing or mitigating the negative effects of economic slowdowns on the HRH, and ultimately increase population access to health services.[17]

## METHODS
This study protocol refers to a systematic review of the existing qualitative, quantitative and mixed-method evidence research, with the aim of building an evidence-based conceptual framework.

This systematic review focuses on generating an evidence-based theoretical framework, strengthening the current understanding of a complex phenomenon with sociopolitical and economic roots, and inform respective health policies, but not assess the effects of specific health interventions. We do not use a traditional *aggregative* or Cochrane-style systematic review templates, but rather a *configurative* systematic review, both combining and synthesising diverse types of knowledge into an overarching framework.[18–22]

Where appropriate, this systematic review protocol was built with reference to the Preferred Reporting Items for Systematic Review and Meta-Analysis Protocols (PRISMA-P)[23]; in some instances, some features of the PRISMA-P were not considered suitable for this review, such as the use of the Participants, Interventions, Comparators, and Outcomes (PICO approach) to set the review question or eligibility criteria.[24] The PICO approach is particularly suitable for a systematic review of the effects of interventions. However, in this review, we rather aim to deepen the knowledge about a wider and complex social phenomenon, not a discrete intervention or set of interventions, and translate that deeper knowledge into an evidence-based conceptual framework.

With the objective of combining quantitative and qualitative information, we will use a 'data-based convergent synthesis design', with all types of data synthesised under the same method[20]; herein, quantitative or mixed-method data will be synthesised qualitatively within thematic categories,[20 21 24] while those categories will be initially derived from an a priori conceptual framework.

Within such rationale, we will apply the 'best-fit framework synthesis' approach to the data synthesis.[24] Framework synthesis approaches are deductive forms of qualitative data synthesis (ie, use a relevant a priori framework against which the reviewed information is coded and synthesised against), and are increasingly used in health systems and policy research, essentially due the theoretical soundness, feasibility and the relative simplicity of the approach and its interpretation.[21 22 25 26] The 'best-fit' framework synthesis is a recent variant of the method,[24] which broadly retains the same advantages but also allows for inductive changes in the underlying framework—notably for accommodating emergent themes from the literature not covered by the initial framework.[22 27] As one limitation, though, the best-fit framework synthesis is an emergent method of literature synthesis whose approach is still evolving and being methodologically refined.[22] Another limitation is the existence of some subjectivity for the research team in the selection of relevant theories or models for the building of the a priori framework, against which the data will be later synthesised.[22] With this respect, and as a partial countermeasure, we will involve

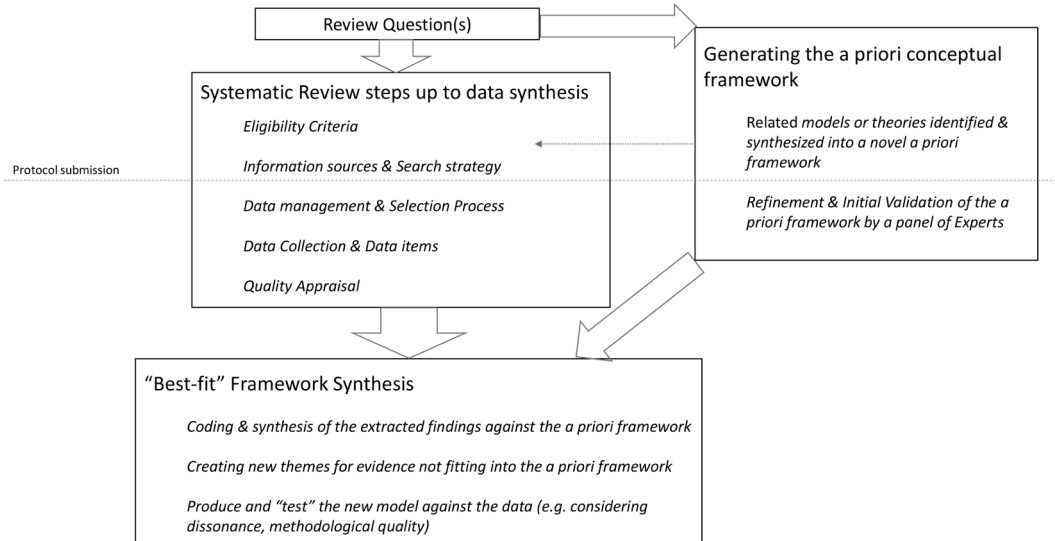

**Figure 1** Flow chart of this systematic review using a 'best-fit' framework synthesis.

a set of experts, not part of the research team and with varying backgrounds, in the refinements of that a priori framework, before the data synthesis.

Along with these lines, figure 1 synthesises the flow of the steps in conducting this systematic review.

### Generating the a priori framework

In addition to the typical systematic review efforts, a best-fit framework synthesis implies generating an a priori framework against which the systematically reviewed information will be coded and synthesised.[22] Despite the growing literature on economic recessions and health systems, the research team (which includes senior scholars with health systems and human resources expertise: EK and GR) is not aware of any framework linking specifically economic recessions to impacts on the HRH. We have therefore built on the mainstream frameworks by economic, public health and health system research scholars exploring the pathways through which an economic recession affects healthcare.[7 28–33]

This initial understanding was confirmed by a further screening of the literature (titles and abstracts) indexed in PubMed, conducted in the early 2019 by one other coauthor (TSJ), using the search strategy depicted in box 1. That search revealed 120 entries, none of which with a framework specifically linking economic recessions to impacts on the HRH. As we could not build an a priori framework though a synthesis of existing ones,[22] we

> **Box 1 Search strategy details in PubMed we used, although unsuccessfully, to try locating any theoretical frameworks specifically addressing the effects of economic recessions on the human resources for health**
>
> ("Models, Theoretical"[Mesh] OR framework* OR Theor* OR Concept*) AND ("Economic Recession"[Mesh] OR "Economic Recession*" OR "financial cris*" OR "austerity") AND ("Health Workforce"[Mesh] OR "Health Personnel"[Mesh])

used an alternative approach relying on and combining key features of the broader economic,[7 8] health systems frameworks[7 28–33] and human resources[7] theories to initially build our a priori framework. Figure 2 depicts our initial, synthesised a priori framework as built by the research team.

Such a priori framework is provisional, and will be further refined, synthesised and/or validated by a panel of external experts. A minimum of three experts (ie, senior scholars), each corresponding to an economic, health systems, and health workforce background, will take part in that refinement and validation process, which will include additional theories and models not previously accounted for by the research team, for instance, any overlooked models and theories arising from the knowledge fields they have an intimate knowledge of. Experts will be identified through snowballing from personal and professional acquaintances of the research authors, and then invited through email, along with the designated roles and the study protocol. Those accepting the role will need to explicitly agree that their names (although not their individual input) to be later highlighted in the final report. A maximum of two iterations (ie, personally, through videoconference, or email exchange) may occur between each of the experts and at least two members of the research team. The iterations may or may not be needed to clarify or further revise the initial input initially and independently provided by each expert, in face of the initial feedback provided by the other experts too. The combined result of the alterations into the framework, from that initially built by the research, will be reported in the final study's report. The new, revised a priori framework will be the one guiding the initial coding and synthesis of the systematically reviewed information. Any alternative formulations (eg, suggested edits to the initial model from different experts that are not aligned with one another) from and among the experts will be kept by the research team to be later tested against the data emerging from the systematic review.

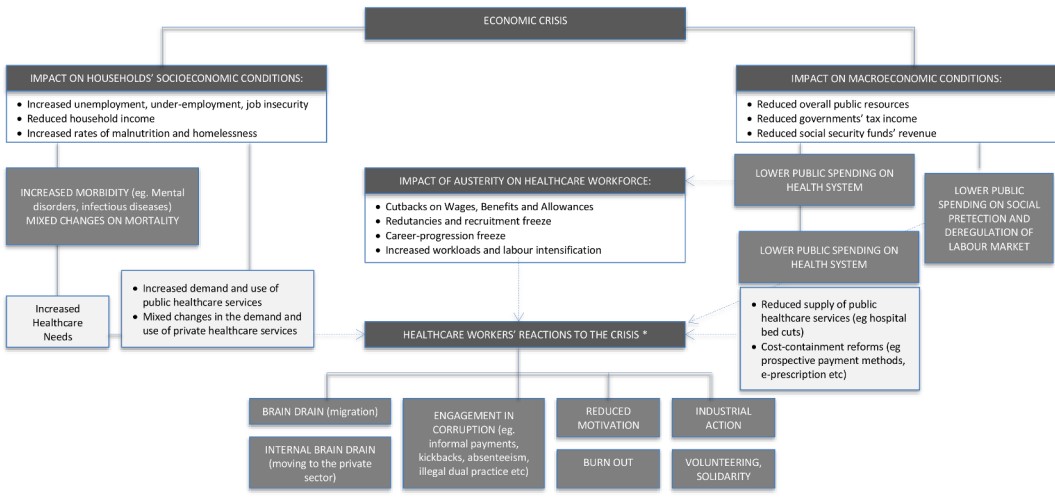

**Figure 2** Theoretical, a priori framework of the impact of economic recessions on the human resources for health.

## Systematic review steps up to data synthesis

Before conducting the data synthesis against the a priori framework, we will carry out the typical systematic review steps, with the following specifications.

### Eligibility criteria

We will include peer-reviewed studies whose aims and results address (ie, explore, determine, investigate) at least one of the following subjects:

1. The impact of economic recessions on the HRH.
2. Any reaction of the HRH to economic recessions.
3. How any of these impacts or reactions have been associated to or affected (ie, result from, were mediated or moderated, prevented or mitigated) by any public policies in response to the economic recessions, such as but not limited to the so-called austerity measures.

For the key terms used, we apply the following conceptual definitions:

▶ Economic recession: Economic slowdowns affecting one or more countries, where gross domestic product contracts for at least two consecutive quarters. Episodes of stagnations, slow growth and less than two quarters contractions were excluded.[34]
▶ HRH: Any clinical health workers (eg, physicians, nurses, allied health professionals, pharmacists, oral health or eye health practitioners, community health workers) working in the health, social care or other (eg, educational, labour) sectors. Clinical researchers are included as they have a key role for and are often part of the national health systems. Clinical residents, fellows, or students, even undergraduates, are also included, as they are part of the either current or prospected workforce. Such cadres are likely to soon enter into the health labour market, affect its dynamics, and be affected by the impact of economic recessions. Non-clinical staff, such as administrative staff, managers, assistants, other technicians and health information staff are not necessarily health specific and excluded from the definition.

If the definitions above prove to be insufficient for reliable selection decisions, further specification might be added, with consensus among the research team, and then made explicit in the final publication.

We will include studies of any design that collect and report results from primary quantitative, qualitative or mixed-method research, from every location, as we aim to review the wealth of evidence coming from different recessions in different settings. Indeed, analysing and contrasting the diverse evidence will be one of the key features of this review. We will also include original analysis of secondary data and systematic reviews, including scoping reviews, realist synthesis or other forms of systematic (ie, reproducible) knowledge synthesis. We will, however, exclude the grey literature (unpublished studies or studies without scientific peer-review, such as preprints) as well as non-systematic reviews (ie, traditional narrative literature reviews, editorials, commentaries, opinion pieces, perspective papers, letters or correspondence).

We will include papers published from 1970 until 2019. The time period of our analysis covers all major global or regional economic recessions that affected both the developed and developing world[35] and it is commonly used in similar studies.[36] We will use no setting restrictions, as studies will be able to address any geographical location or set of locations. We will exclude papers with no title and/or abstract available in English or which the full text is not in English, French, Greek, Italian, Spanish or Portuguese. These are languages the research team can handle, in which the bulk of the peer-reviewed literature and studies are reported, and include languages from countries more recently affected by economic slowdowns or subject to austerity measures (eg, Angola, Brazil, Greece, Ireland, Italy, Mozambique or Portugal). Whenever papers are in languages in which the research team has less than two fluent readers (eg, in Greek, Italian), full texts might be fully translated into another language (ie, English), a priori of the full-text review, either by a

---

**Box 2  Search strategy details in PubMed to locate relevant papers for this systematic review**

("Economic Recession"[Mesh] OR "Economic recession*" OR "economic slowdown*" OR "economic slow-down*" OR "spend* slowdown*" OR "spend* slow-down*" OR "economic cris*" OR "economic contraction*" OR "financial cris*" OR "austerity") AND ("Health Workforce"[Mesh] OR "Health Personnel"[Mesh] OR "Employment"[Mesh] OR "Burnout, Professional"[Mesh] OR "Emigration and Immigration"[Mesh] OR "Career Choice"[Mesh] OR "Personnel Staffing and Scheduling"[Mesh] OR "dual practice*" OR "brain drain*" OR "absenteeism*" OR "corrupt*" OR "recruit*" OR "retention*" NOT "Child Labor"[Mesh] NOT "Employment, Supported"[Mesh]) AND ("1970/01/01"[PDAT] : "3000/12/31"[PDAT] AND (English[lang] OR French[lang] OR Greek, Modern[lang] OR Italian[lang] OR Portuguese[lang] OR Spanish[lang]) NOT (Letter[ptyp] OR Editorial[ptyp] OR Comment[sb]))

---

native speaker within the research team or by an external professional translator.

### Information sources and search strategy

Towards identifying relevant papers, we will first approach eight scientific databases (PubMed; ISI Web of Science—core collection; Scopus; Cochrane Library; PDQ-Evidence; Health Evidence.org; Scielo and Econ Lit), from 1970 until current date. Altogether, these databases are known to cover the mainstream health, public health, health systems, multidisciplinary and economic literature.

Informed by our initial a priori framework and our eligibility criteria, we have defined a first full search strategy in PubMed, which was additionally checked against the guidelines of the Peer Review of Electronic Search Strategies for systematic reviews.[37] The search strategy in box 2 was designed by one of the researchers (TSJ) with track record of designing search methods and strategies,[38–43] also supported by the study questions and the preliminary model shown in figure 2. The same author will conduct the searches for the other databases.

In addition to databases searches, one of the lead authors (GR) will conduct snowballing searches (ie, citations tracking; authors tracking; references list consultation) over the articles finally selected through the databases searches, that is, looking for any additional relevant references. Such secondary searches will be run only at a later stage once the selection of references coming from database searches have been completed. Secondary searches also entail searching websites and/or databases (eg, using key search terms) of the following institutions: European Observatory on Health Systems and Policies; the World Bank, and the Organisation for Economic Co-operation and Development. Although we do not include papers from the grey literature, the searches on the institutional websites can help identify relevant empirical, peer-reviewed material, for example, cited into key reports from the institutional sources.

We will export databases searches to the Endnote software, which will be used to deduplicate entries and to produce tailored Excel spreadsheets with the titles and abstracts for the level 1 screening. That level 1 screening will be fully performed by one researcher with review experience (TSJ) after a pilot test with a random sample of entries (5% of the total).

That pilot testing will consist on one senior scholar with subject matter expertise (GR) verifying all the exclusions the tasked reviewer made against the eligibility criteria. Dubious or discording cases will trigger further clarification/specification of the eligibility criteria—and then a new pilot test. The pilot test will be run as many times needed up to an agreement of the lead author with all the exclusion decisions made by the tasked reviewer, within a 5% random sample of entries. After such an agreement, the tasked reviewer will proceed with the level 1 screening. Finally, even in the full level 1 screening, the tasked reviewer will mark any potentially dubious eligibility decisions for verification by the senior scholar (GR) before a definitive elimination takes place.

For the level 2 screening (ie, full-text review), two reviewers (eg, GR being reviewer number 1 throughout; JF and EK splitting the role of the reviewer number 2) will independently determine the eligibility of the papers. If a large amount of full texts is to be reviewed, we may include additional persons (eg, new research authors) towards performing the reviewer number 1 and number 2 roles, and/or redistribute the initially assigned tasks. Any new person will be trained or supervised accordingly by initial member of the research team. Also, whenever full texts are in languages other than English, preference will be given for using the reviewers more fluent in the respective language, among those available/trained for the task. Whenever, reviewer number 1 and number 2 agree, the paper is immediately included or excluded. When they initially disagree, the two reviewers will have the opportunity to discuss their rationales up to a consensus, with a third reviewer being called whenever needed.

The research team will construct a data extraction table containing the following items: study design (eg, economic evaluation; cross-sectional analysis; case report)—per the 12 different types of study designs (excluding opinion pieces) from the list of critical appraisal checklists of the Joanna Briggs Institute[44]; type of data (eg, qualitative, quantitative, mixed) sampling procedures; sample characteristics (sample size, geographic area and demographic characteristics); study design (including sampling procedures); data source (for analyses based on previous surveys); outcome(s) variables; methods used in analysis; and variables controlled for. A preplanned coding structure will be developed for each of the subjects above. For all these data items, one reviewer will extract and code the data (FJ)—asking for verification of a second author (GR) whenever the coding option is not straightforward (ie, when interpretation is involved, or any subjectivity may arise).

The data extraction for the main findings, which consist of either quantitative or qualitative data that could directly inform the refinement of the conceptual

model, will be conducted in a different manner. There will be no coding structure, essentially as we want the evidence-based data, arising from the different methodologies included and subject matters addressed, to inductively emerge from the included papers before being analysed and synthesised against the a priori framework. Two independent reviewers (those conducting the level 2 screening) will extract data, which will be then merged and deduplicated.

Authors of any included papers may be approached by the research team, via publicly disclosed email contacts, for clarification or to provide additional data on filling for the data extraction tables, when there is key information that has not been explicitly reported within the included papers.

### Quality appraisal

Each publication finally selected through the level 2 screening will be appraised for methodological quality. We will use the tools appropriate for the study design, from the entire portfolio of the Joanne Briggs Institute's critical appraisal tools, covering 12 different types of study designs.[44] Two independent reviewers (those with the level 2 screening and extracting tasks) will apply the respective checklist, according to the previous study-design classification.

Within that process, at the end of the critical appraisal, each reviewer will preliminary recommend the 'inclusion', 'exclusion' or the option to 'seek further information', according to the methods quality. However, as typical in exploratory reviews or those on complex issues, papers with methodological shortcomings (ie, those recommended to be excluded) may not necessarily be excluded immediately (ie, as a prequalification exercise), but eventually at the later synthesis stage—for example, using well-known qualitative saturation principles (ie, when there is no additional or qualitatively different information from a source with lower methodological quality).[24 45] Evidence coming from a paper with relevant methodological shortcomings will be signalled as such in the paper's final report.

### Best-fit framework synthesis

The data (ie, quantitative and qualitative findings) extracted from the papers will be at this stage coded and synthesised following the a priori framework. Qualitative and quantitative evidence will be analysed together using the same synthesis method,[20] that is, within the qualitative, thematic categories of the a priori framework. That synthesis process will be initiated by the one research author (GR), and then iteratively edited by the whole research team.

Should the evidence from the systematic review not fit into thematic categories of the a priori framework, new thematic categories, subcategories or linking mechanisms might be created as emerging from the data. This will allow for a refined, more nuanced

conceptual framework that is not only theoretically but also empirically sound.[22]

During the synthesis, the emergent conceptual model will be tested for sensitivity. Indeed, within a best-fit framework synthesis, authors need to determine if the synthesis is sensitive to variables such as the reported quality, design or location of included studies.[22] For example, we will test (ie, compare) the effects on the model regarding the use of evidence arising only from high-income countries or from LMICs. This may imply adjustments to the model for it to be sensitive or adaptive to varying locations or economic contexts. Similarly, we will the test the model for sensitivity regarding the inclusion or exclusion of evidence with methodological shortcomings. Along with qualitative saturation principles, such methodological shortcomings will help determine whether those references, findings or new thematic categories will be included or excluded in part or as a whole. However, this will apply only to the papers that are not 'fatally flawed'.[24] Studies with methodological shortcomings, or areas for which studies do not exist or provided divergent findings also might hint for further research.

Should there be significant differences for including evidence with lower quality assessment, both models will be presented for reader's assessment. Differences to the underlying frameworks may also be further explored through the search for dissonant or negative cases.[22] The effects of considering only particular types of studies or variables will also be explored, as a way to strengthen the model or help identify context-sensitive nuances and implications.

Finally, any alternative configurations of the a priori framework resulting from differences of opinion from the experts' input will be further explored and validated against the data.

### Patient and public involvement

This systematic review will involve external experts (ie, senior scholars not part of the research team and with background expertise in economic, health systems and/or health workforce issues) in shaping the a priori framework that will guide the data synthesis. Despite not including direct input from health workers (ie, those addressed by the study aims), this review aims to include papers with original data about how economic recessions directly and indirectly impact on health workers, including from their own perspectives.

## ETHICS AND DISSEMINATION

This protocol refers to a systematic review and thereby does not involve primary data collection. Experts will be approached either through personal networks or through public-domain email contacts. They will be able to refuse involvement and, in that case, their names will not appear in the final study's report.

Those accepting the oversee role will need to explicitly consent their names and positions to be mentioned; however, their individual contribution will only be revealed in the final outcome of the interaction with the authors of the review.

We intend to publish the final study's report in a peer-reviewed journal, along with depositing the underlying review data (eg, data extraction forms, level 2 screening sheets) within a public, online repositorium for added transparency.

The dissemination strategy for the review's results will include presentations at regional as well as global conferences. An example would be that of the Health System research Symposium.

Finally, the results of this systematic review (ie, the evidence-based conceptual framework) will be used by a research author (GR) for regional policy dialogue workshops in Brazil. Those workshops are part of a funded research project focusing on the impact of the economic recessions in Brazil and on the available policy options to mitigate its effects for health system and health workers, funded by the UK Medical Research Council and the Newton Fund.

**Author affiliations**
¹Global Health and Tropical Medicine, WHO Collaborating Center on Health Workforce Policy and Planning, New University of Lisbon, Institute of Hygiene and Tropical Medicine, Lisbon, Portugal
²Centre for Primary Care and Public Health, Queen Mary University of London, London, UK

**Contributors** All authors were involved in defining the concept, design and overall definitions of the study protocol. TSJ has translated the team definitions into the details of the search strategy in the scientific databases and has written the first draft of the protocol methods. GR, who is the guarantor of the review, has written the draft introduction of the study protocol and the draft dissemination strategy. All authors have iteratively revised the draft protocol and agreed with its final definitions.

**Funding** GR was supported by the Medical Research Council and Newton (Fund grant number MR/R022747/1). The funder had no role in the development of the protocol.

**Competing interests** None declared.

**Patient consent for publication** Not required.

**Provenance and peer review** Not commissioned; externally peer reviewed.

**ORCID iDs**
Tiago Silva Jesus http://orcid.org/0000-0003-1300-6308
Elias Kondilis http://orcid.org/0000-0001-9592-2830
Jonathan Filippon http://orcid.org/0000-0003-3907-1992

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
