## [Reviewer comments · BMJ Open]

ARTICLE DETAILS

TITLE (PROVISIONAL)	The impact of economic recessions on healthcare workers and their crises' responses. A study protocol for a systematic review of the qualitative and quantitative evidence for the development of an evidence-based conceptual framework
AUTHORS	Jesus, Tiago; Kondilis, Elias; Filippon, Jonathan; Russo, Giuliano

VERSION 1 – REVIEW

REVIEWER	Eleni Koutsogeorgou University of Sussex United Kingdom
REVIEW RETURNED	14-Aug-2019

GENERAL COMMENTS	General comments: • This paper attempts to present a protocol for systematic review relevant to economic recessions' impact on health workers. The topic is interesting, but the basis for the methodology proposed is not clear.• The main issues are that the terminology used is not clear and the aim of the study is not clearly defined either.• Authors claim that they can develop a conceptual framework for analysis of economic recessions' impact on health workers or their reactions during economic recessions (not clear). However, the type of the impact explored is not presented, nor supported by relevant literature in the introductory parts, which is also obvious in the search terms used that are not defined or justified by authors for appropriateness in this review.• It is not clear how the methods that will be identified in existing scientific literature will be combined and used to create a newly developed conceptual framework. The authors need to clarify each step of the process, considering the barriers of language, large variety of contexts from the studies that will emerge, and the limitations of their study due to the lack of search terms at the initial phase of the database search.• Authors should use the aim(s) of their study to guide the methodology of the review. These do not appear as closely connected and explored adequately with the use of the methodology presented in the manuscript. For example, how will the impact on HRH or their 'reactions' will be measured? How will 'key impacts' of recession be identified in this review? It seems that data extraction will mainly focus on the methods used in studies and less on the findings relevant to the impact of recession on HRH. Specific comments:
---

	Page 2, line 33: Should present the extended form of acronyms the first they are mentioned in the text (i.e. 'HRH'). Page 2, line 42: "will be searched" instead of "by". Page 2, line 51: If two reviewers assess the full text and the language of articles included is Portuguese, for example, this means that there are at least two researchers who speak Portuguese fluently and can assess the text. In other words, at least two researchers can speak these languages (i.e. Spanish, Italian, French, Portuguese or Greek) in order to be able to assess the suitability of the text for inclusion. If so, this should be stated in the Methodology. Page 3, line 12: Why 'Latin America' only? Page 3, line 46: Correct "an evidence-based". Page 5, line 3: Why would it be expected to have 'public involvement' for this systematic review? Page 7, line 31: It is too vague to claim that you can evaluate how HRH 'have reacted' in economic recessions. It is expected that health systems and governments react to economic recessions by reducing public expenses and possibly funds for HRH, so maybe should evaluate how health systems/governments reacted regarding HRH and not how HRH reacted per se? Page 7, line 41: Could use a citation to argue that population access to health services depends on changes in HRH. Page 8, line 38: Why PICO was not appropriate for this review? Page 10, line 10: Models or theories relevant to what? Health and healthcare? The aim of the study is on HRH so why health-related models might be of interest here? How these will be identified before the review at design level? What do these theories or theories focus on? Who selects them? It is not novel to conduct a systematic review using an a priori coding scheme or framework (based on models or theories or the main research questions) and extract data during synthesis of findings which are relevant to this framework. Thus, it is not clear how this 'best fit' framework differs from previous review studies. Page 11, line 5: Why only 'Pubmed'? Page 11, line 10: Not clear what 'defined framework' entails. Page 11, line 31: Please clarify what "additional theories and models" refers to. Page 11, line 42: Not clear what the purpose of 'iterations' is. Page 11, line 54: What 'alternative formulations' refer to? Page 13, line 11: No need to put in a Box the combination of search terms, can be inserted as a phrase within the text. Why did you use plural for 'models' and not 'model*' or 'economic cris*' as search terms? Possibly could use in search terms also categories
--	--

	of HRH such as doctors, nurses, health professionals, etc. in the search terms too? It is not clear why ‘theoretical’ and ‘models’ were searched, while there are many publications on the effects of economic recessions on health systems including HRH, which might have been informative for this study. Page 13, line 50: Again, these two subjects for the review are too vague. Page 14, line 3: It is not clear how else the impact of economic recession on HRH would occur without policies and as a result of the economic recession itself. Page 14, line 27: Here HRH is stated that is a key term that includes a variety of professions (e.g. clinical health workers, clinical researchers, etc.), which however were not included as search terms in the review. Why administrative and managerial staff were excluded from the definition of HRH since, apparently, they have an important role in the health system and could potentially have influence patient satisfaction and/or access to healthcare services. Page 16, line 32: Again, the search terms used for health workers are not exhaustive, could have used more search terms such as specific roles, ‘health workers’, ‘health staff’, ‘doctors’, ‘physicians’, ‘nurses’, etc. There might be a study, for example, on the impact of economic recession on nurses, so this study would not have been included in the findings of this database search with these terms presented in the manuscript. Page 16, line 34: Why is the search term “Emigration and Immigration” relevant? If relevant, then why the two terms were combined together only and not separately in the search? Not clear why other terms (such as ‘absenteeism’, ‘dual practice’, ‘corrupt’, etc.) were relevant and included in the search terms. There has to be justification backed up with scientific literature for the choice of search terms. Page 17, line 10: It is not clear why authors would search these institutions/organisations’ websites, since these institutions mainly publish technical reports, or else grey literature – which authors stated above that would be excluded from this review. Page 18, line 19: How is the ‘data extraction table’ different from the framework that would be developed as described previously in the study? Why this ‘data extraction table’ does not include any relevant models or theories potentially presented within the selected articles? When will these be extracted during the review, if not during the screening phase of full texts? Page 20, line 27: It is not clear what is meant by ‘the emergent conceptual model will be tested for sensitivity’ or how this will be performed.
--	---

REVIEWER	Ana Antunes Comprehensive Health Research Centre (CHRC) NOVA Medical School, Nova University of Lisbon
REVIEW RETURNED	03-Oct-2019

GENERAL COMMENTS	Dear authors, I appreciate the opportunity to review this protocol, which aims to fill an important gap in the literature by providing a conceptual framework for the impact of economic recessions on human resources for health. I recommend the publication of the protocol and I have just some minor suggestions for the authors: Abstract: I suggest the authors to add "human resources for health" before the respective acronym (line 34); Main text: - I suggest the author to standardize the terms economic recession/contraction/crisis (e.g. line 82; line 157) - I suggest the authors to re-write the sentence on line 96-97 due to repetition of terms; - Please clarify the following paragraph: line 301-304; - If appropriate, a brief reflection on the possible shortcomings/limitations of the best-fit framework synthesis in the context of this study could enrich the manuscript ; Lastly, I would also like to suggest the authors to check the following publication: Kentikelenis, A.E., 2017. Structural adjustment and health: A conceptual framework and evidence on pathways. Social Science & Medicine, 187, pp.296-305. This conceptual framework refers to the direct effects of austerity measures on the health system, focusing on the healthcare workforce, and may be useful for your work. Best of luck with your study.
---

VERSION 1 – AUTHOR RESPONSE

Reviewer 1's comments

1. General comments: This paper attempts to present a protocol for systematic review relevant to economic recessions' impact on health workers. The topic is interesting, but the basis for the methodology proposed is not clear. The main issues are that the terminology used is not clear and the aim of the study is not clearly defined either.

The aim of the study is to review the evidence on the impacts of economic recessions on HRH, and use it to draw a framework for the conceptualization of the repercussions. We have now clarified our aims at the end of the Introduction (new content underlined):

“The aim of the study is to review the evidence on the impacts of economic recessions on HRH, and use it to draw a framework for the conceptualization of the repercussions. We aim to do so through a synthesis of both the qualitative and quantitative evidence of the impact of economic recessions on the HRH. In particular, we are interested in reviewing qualitative and quantitative evidence on: 1) any direct or indirect impacts of economic recessions on the HRH (salary cuts, loss of motivation, reorganization of provision of services etc.), including identifying any mediating or moderating variables, and the 2) health workers' reactions to such effects (migration, engagement in alternative profit generating activities, early retirement, etc). The evidence arising from each subject will be combined into a single, yet comprehensive, evidence-based conceptual framework of the conjunct of effects of economic recessions on the HRH.

In that context, our specific study questions are:

A. What have been the key impacts of economic recessions, occurred since 1970, for the global and national health workforces, and through which mechanisms, mediating or moderating variables have those impacts occurred?

B. How have HRH in different contexts reacted (i.e. emigrated, became involved into dual practice, etc) to the consequences of economic recessions, or to any resultant policy measures, such as but not limited to the so called austerity measures?"

It is hoped that the framework from the uncovered evidence will strengthen the current understanding of this complex socio, political, economic and labour market phenomenon toward informing both ex ante and ex post policies on either preventing or mitigating the negative effects of economic slowdowns on the HRH, and ultimately increase population access to health services.¹⁷

Later in the methods section, we provide a key figure (i.e. figure 2) with an a priori framework showing possible moderating or mediating variables and mechanisms involved in the direct or indirect impact of economic recessions on the HRH – per our juxtaposition of relevant economic, health system and human resources theory. Please note that, as explained in the main manuscript, that is only a provisional framework that will be subject to further refinement by an external group of experts and, then, more importantly, by the evidence reviewed – which will determine the final shape of the model (i.e. the final review's output).

Finally, in the methods section, we provide conceptual definitions of the two key concepts. The definitions are transcribed below:

"For the key terms used, we apply the following conceptual definitions:

- Economic recession: Economic slowdowns affecting one or more countries, where Gross Domestic Product (GDP) contracts for at least two consecutive quarters. Episodes of staginations, slow growth and less than 2 quarters contractions were excluded.³⁴
- HRH: Any clinical health workers (e.g. physicians, nurses, allied health professionals, pharmacists, oral health or eye health practitioners, community health workers) working in the health, social care or other (e.g. educational, labour) sectors. Clinical researchers are included as they have a key role for and are often part of the national health systems. Clinical residents, fellows, or students, even undergraduates, are also included, as they are part of the either current or prospected workforce. Such cadres are likely to soon enter into the health labour market, affect its dynamics, and be affected by the impact of economic recessions. Non-clinical staff, such as administrative staff, managers, assistants, other technicians, and health information staff are not necessarily health-specific and excluded from the definition. "

All the above material provided in the methods section, rather than in the Introduction, as these are used as working definitions – i.e. designed to ground the reviewers' decisions on selection or not relevant articles; hence more appropriate for the methods section, although otherwise also relevant for earlier parts of the manuscript.

2. Authors claim that they can develop a conceptual framework for analysis of economic recessions' impact on health workers or their reactions during economic recessions (not clear). However, the type of the impact explored is not presented, nor supported by relevant literature in the introductory parts, which is also obvious in the search terms used that are not defined or justified by authors for appropriateness in this review.

We have further clarified the aim of the review - see above.

In the Introduction, we have explored, in part, the types of impacts of economic recessions in the HRH, supported by the literature, as transcribed below:

"As other workers during an economic recession, healthcare workers are affected by a combination of job insecurity, decreased purchasing power and reduced labour market opportunities.⁹ It has also been suggested that these effects are often amplified by the reduction of welfare support and salary cuts by the restrictive policies governments typically apply to the largest spending sectors (including commonly health) in the attempt to balance budgets and reduce deficits.^{8 10}

The known effects of economic contractions and austerity policies on health workers include salary cuts and job losses, compounding migration intentions,^{11 12} decreased motivation,¹⁰ unwanted organizational changes,¹³ and an increase in the tendency to engage in concurring profit generating activities,¹⁴ often at the expense of the quality of service. Scholars have started to explore the area of the effect of Europe's most recent financial crisis on human resources for health (HRH) policies,¹⁵ and health workers' responses to the changing economic circumstances,^{10 11 16}

However, we also acknowledge we have not explored these impacts exhaustively in the Introduction, as we believe the explanations around Fig.2 in the methods accomplish such task in a more comprehensive way (i.e. with visual display of the variables and linking mechanisms), with the support of the existing theories, as supported in the respective parts of the text (transcribed below).

"We have therefore built on the mainstream frameworks by economic, public health and health system research scholars exploring the pathways through which an economic recession affects healthcare.^{7 28-33} (...) "relying on and combining key features of the broader economic,^{7 8} health systems frameworks^{7 28-33} and human resources⁷ theories to initially build our a priori framework. Figure 2 depicts our initial, synthesized a priori framework as built by the research team."

With respect to search terms, we acknowledge they may appear limited on a superficial examination; however, most of the terms we used were MeSH terms (not free-text keywords) and, by default in PubMed search, inclusive of several other MeSH terms within (all of those below, within the hierarchical tree of MeSH terms). For example, "Health Personnel"[MeSH] is inclusive of "Nurses"[MeSH], "physicians" [MeSH], etc. Even though we don't write the latter down – yet, by including "Health Personnel"[MeSH], they are included, as well. The same applies to several other aspects of our search strategy, which seems less comprehensive than it technically is. We respond below with more technical details about the search terms – within this reviewers' specific comments regarding their adequacy.

Lastly, the reviewer is right with regards to the justification of the search terms. Indeed, we did not report our grounds before, notably when presenting the main strategy. But we do so now (new content underlined):

"The search strategy in Box 2 was designed by one of the researchers (TJ) with track record of designing search methods and strategies,³⁸⁻⁴³ also supported by the study questions and the preliminary model shown in the figure 2."

3. It is not clear how the methods that will be identified in existing scientific literature will be combined and used to create a newly developed conceptual framework. The authors need to clarify each step of the process, considering the barriers of language, large variety of contexts from the studies that will emerge, and the limitations of their study due to the lack of search terms at the initial phase of the database search.

Our review will consider quantitative as well as qualitative evidence. Please note we are not planning to undertake a traditional review focusing only on one type of evidence or studies, but rather reviewing the wealth of evidence coming from different methods and addressing different economic recessions in different settings. Analyzing and contrasting the diverse evidence will be, we believe, one of the key strengths of our study, as it will enable our framework to capture the context-specific nature of different impacts and behaviors related. Indeed, to clarify this point, in the Methods section - under the eligibility criteria, we now report the following (new content underlined)

"We will include studies of any design that collect and report results from primary quantitative, qualitative or mixed-methods research, from every location, as we aim to review the wealth of evidence coming from different recessions in different settings. Indeed, analyzing and contrasting the diverse evidence will be one of the key features of this review.

Then, at the synthesis stage, we also now report the following (new content underlined):

"During the synthesis, the emergent conceptual model will be tested for sensitivity. Indeed, within a "best-fit" framework synthesis, authors need to determine if the synthesis is sensitive to variables such as the reported quality, design or location of included studies.²² For example, we will test (i.e. compare) the effects on the model regarding the use of evidence arising only from high-income

countries or from low- and middle-income countries. This may imply adjustments to the model for it to be sensitive or adaptive to varying locations or economic contexts.”

4. Authors should use the aim(s) of their study to guide the methodology of the review. These do not appear as closely connected and explored adequately with the use of the methodology presented in the manuscript. For example, how will the impact on HRH or their ‘reactions’ will be measured? How will ‘key impacts’ of recession be identified in this review? It seems that data extraction will mainly focus on the methods used in studies and less on the findings relevant to the impact of recession on HRH.

The data extraction will be focused on both the methods used and the findings relevant to the impact of recession on HRH. The data extraction on the latter is, however, implemented in a different way – i.e. not with a pre-defined coding scheme, given the diversity of the studies and methodologies (as noted above) under review, some of which (e.g. qualitative studies) may not necessarily use of “measure” of impact. Hence, in the “data extraction” section, we now further clarify how the data extraction is going to be performed regarding the findings relevant to the impact of economic recession on HRH (all the transcribed content is new):

“The data extraction for the main findings, which consist of either quantitative or qualitative data that could directly inform the refinement of the conceptual model, will be conducted in a different manner. There will be no coding structure, essentially as we want the evidence-based data, arising from the different methodologies included and subject matters addressed, to inductively emerge from the included papers before being analysed and synthesized against the a priori framework.”

5. Specific comments: Page 2, line 33: Should present the extended form of acronyms the first they are mentioned in the text (i.e. ‘HRH’).

We have now spelled the acronyms when they first appear in the text.

6. Page 2, line 42: “will be searched” instead of “by”.

Thanks. This has now been corrected.

7. Page 2, line 51: If two reviewers assess the full text and the language of articles included is Portuguese, for example, this means that there are at least two researchers who speak Portuguese fluently and can assess the text. In other words, at least two researchers can speak these languages (i.e. Spanish, Italian, French, Portuguese or Greek) in order to be able to assess the suitability of the text for inclusion. If so, this should be stated in the Methodology.

For the case of Portuguese, indeed we have 3 research authors that are fluent in the language, including two native speakers. Though, there are languages in which that does not happen. And, indeed, we failed to report how we plan to proceed with these regards. We have now specified the following in the main manuscript:

“Whenever papers are in languages in which the research team has less than 2 fluent readers (e.g. in Greek, Italian), full texts might be fully translated into another language (i.e. English), a priori of the full-text review, either by a native speaker within the research team or by an external professional translator.”

We also took the opportunity to add additional details in the main manuscript regarding the assignment of the reviewer roles, including how is that flexible to accommodate papers in languages other than English. So, when we mention how the full-text review will be conducted, we now also state the following:

“If a large amount of full texts is to be reviewed, we may include additional persons (e.g. new research authors) toward performing the reviewer number 1 and number 2 roles, and/or redistribute the initially assigned tasks. Any new person will be trained or supervised accordingly by initial member of the research team. Also, whenever full-texts are in languages other than English, preference will be given for using the reviewers more fluent in the respective language, among those available/trained for the task.”

8. Page 3, line 12: Why 'Latin America' only?

Latin America will be the focus of one specific dissemination event planned in the main research project: "Regional policy dialogue workshops". In the main manuscript, i.e. its last paragraph, we highlight the specific reasons for the that context to be addressed by that dissemination event in particular. Unfortunately, it is not feasible to apply regional police dialogues to more regional contexts. We report, though, in the abstract, two additional dissemination events with wider, geographical outreach: i.e. peer-review publication and presentation at global health systems conferences. Crucially, following this comment from the reviewer, we recognise that the dissemination strategy focused in global health systems conferences was depicted only in the abstract and, by our mistake, not in the main manuscript. We take the opportunity to correct that, and state now the use of global health systems conferences in the full text. Therefore, in the second to last paragraph of the main manuscript we state the following:

"The dissemination strategy for the review's results will include presentations at regional as well as global conferences. An example would be that of the Health System research Symposium."

9. Page 3, line 46: Correct "an evidence-based".

This has now been corrected.

10. Page 5, line 3: Why would it be expected to have 'public involvement' for this systematic review?

As explained at the very end of the manuscript, this review is part of a wider project aimed at influencing policy-makers in Brazil on the most appropriate policy response to economic recessions. Furthermore, public involvement in research, including systematic reviews, is increasingly expected overall, and a statement about required by the BMJ Open in particular. We added details in the manuscript on this subject, in response to an editorial request - as mentioned above.

11. Page 7, line 31: It is too vague to claim that you can evaluate how HRH 'have reacted' in economic recessions. It is expected that health systems and governments react to economic recessions by reducing public expenses and possibly funds for HRH, so maybe should evaluate how health systems/governments reacted regarding HRH and not how HRH reacted per se?

We agree with this the insightful comment. To this respect, we have changed the way we wrote the paper aims accordingly (new content underlined).

"In particular, we are interested in reviewing qualitative and quantitative evidence on: 1) any direct or indirect impacts of economic recessions on the HRH (salary cuts, loss of motivation, reorganization of provision of services etc.), including identifying any mediating or moderating variables, and the 2) health workers' reactions to such effects (migration, engagement in alternative profit generating activities, early retirement, etc). The evidence arising from each subject will be combined into a single, yet comprehensive, evidence-based conceptual framework of the conjunct of effects of economic recessions on the HRH.

We agree that HRH react to the government's policy measures taken as a response to economic recessions. We have now tried to accommodate that indirect effect, as also highlighted in our a priori model (figure 2).

In figure 2 have tried now to be more specific about the multiple and often indirect pathways by which the economic recessions impact on the HRH and which are their reactions. We understand the figure can demonstrate better all those possible pathways, mediating variables, indirect mechanism, etc. that one could possibly do mostly with text, taking advantage of visual display, arrows, etc.

12. Page 7, line 41: Could use a citation to argue that population access to health services depends on changes in HRH.

Yes, we agree the statement should have bibliographic support. We have now added a appropriate reference from the Bull of WHO:

Campbell J, Buchan J, Cometto G, David B, Dussault G, Fogstad H, Fronteira I, Lozano R, Nyongoro F, Pablos-Méndez A, Quain EE, Starrs A, Tangcharoensathien V. Human resources for health and universal health coverage: fostering equity and effective coverage. Bull World Health Organ. 2013 Nov 1;91(11):853-63. doi: 10.2471/BLT.13.118729.

13. Page 8, line 38: Why PICO was not appropriate for this review?

The PICO approach (Participants; Interventions, Comparators, and Outcomes) is more appropriate for systematic reviews testing the effectiveness of interventions (i.e. the “I” of the PICO). Though, we aim to review a wider social phenomenon. Also, our aim is more exploratory in nature, up to building a conceptual framework to add to the understanding of the complex social phenomenon (a configurative systematic review as we called, with bibliographic support in the antecedent paragraph) while the PICO approach is particularly suitable for an aggregative review, also note in the antecedent paragraph in the manuscript.

We acknowledge, though, we could be more explicit in the paragraph where we specifically mention the PICO was not appropriate here. Hence, we do it now with a new sentence transcribed below: “The PICO approach is particularly suitable for a systematic review of the effects of interventions. However, in this review we rather aim to deepen the knowledge about a wider and complex social phenomenon, not a discrete intervention or set of interventions, and translate that deeper knowledge into an evidence-based conceptual framework.”

14. Page 10, line 10: Models or theories relevant to what? Health and healthcare? The aim of the study is on HRH so why health-related models might be of interest here? How these will be identified before the review at design level? What do these theories or theories focus on? Who selects them? It is not novel to conduct a systematic review using an a priori coding scheme or framework (based on models or theories or the main research questions) and extract data during synthesis of findings which are relevant to this framework. Thus, it is not clear how this ‘best fit’ framework differs from previous review studies.

We agree that it is not novel to conduct a systematic review using an a priori framework. However, the original aim of our best-fit-approach is to build a new review methodology, rather applying an existing review model (i.e. the “best-fit” framework synthesis) to this subject matter and to these review aims in particular. But, indeed, the “best-fit” framework synthesis has weaknesses as a method. So, in response to this whole comment, as well as a similar comment from the reviewer 2, we now state the following in the initial paragraphs of the Methods section (new content underlined):

“The “best-fit” framework synthesis is a recent variant of the method,²⁴ which broadly retains the same advantages but also allows for inductive changes in the underlying framework – notably for accommodating emergent themes from the literature not covered by the initial framework.^{22 27} As one limitation, though, the “best-fit” framework synthesis is an emergent method of literature synthesis whose approach is still evolving and being methodologically refined.²² Another limitation is the existence of some subjectivity for the research team in the selection of relevant theories or models for the building of the a priori framework, against which the data will be later synthesized.²² With this respect, and as a partial counter-measure, we will involve a set of experts, not part of the research team and with varying backgrounds, in the refinements of that a priori framework, before the data synthesis.”

This newly included content aims to complement others that were there before (i.e. in the “1-Generating the a priori framework”), when we mention that:

“We have therefore built on the mainstream frameworks by economic, public health and health system research scholars exploring the pathways through which an economic crisis affects healthcare.^{7 26-31}”

And later (two paragraphs below), we also state that:

“Such a priori framework is provisional, and will be further refined, synthesized and/or validated by a panel of external experts. A minimum of three experts (i.e. senior scholars), each corresponding to an

economic, health systems, and health workforce background, will take part in that refinement and validation process, which will include additional theories and models not previously accounted for by the research team.”

Finally, we strongly agree with the reviewer that our focus, and that of the underlying frameworks, is essentially on healthcare, not health. Hence, we now have deleted the “health” component of the respective sentence (content underlined was deleted).

“We have therefore built on the mainstream frameworks by economic, public health and health system research scholars exploring the pathways through which an economic crisis affects health and healthcare.⁷ 26-31”

15. Page 11, line 5: Why only ‘Pubmed’?

We only used PubMed for a pilot search, but we will use a total of 8 databases for the fully-fledged search process. We selected PubMed given its comprehensive indexation system, with MeSH terms from a hierarchical tree, and by the comprehensive*, albeit not exhaustive, coverage of the healthcare literature.

*Halladay CW, Trikalinos TA, Schmid IT, et al. Using data sources beyond PubMed has a modest impact on the results of systematic reviews of therapeutic interventions. *J Clin Epidemiol* 2015;68:1076-84.

16. Page 11, line 10: Not clear what ‘defined framework’ entails.

We thank you the reviewer for spotting this: the term “defined” does not have a specific role in the statement; it has now been deleted.

17. Page 11, line 31: Please clarify what “additional theories and models” refers to.

The full sentence now states the following (new content addressing the issue is underlined):
“A minimum of three experts (i.e. senior scholars), each corresponding to an economic, health systems, and health workforce background, will take part in that refinement and validation process, which can be grounded into additional theories and models not previously accounted for by the research team; for instance, any overlooked models and theories arising from the knowledge fields they have an intimate knowledge of”.

18. Page 11, line 42: Not clear what the purpose of ‘iterations’ is.

Agreed. Now, in the respective setting, we further state the following:

“The iterations may or may not be needed to clarify or further revise the initial input initially and independently provided by each of experts, in face of the initial feedback provided by the other experts too.”

19. Page 11, line 54: What ‘alternative formulations’ refer to?

We agree we need to clarify. Now, we also state the following:

“(e.g. suggested edits to the initial model from different experts that are not aligned with one another)”.

20. Page 13, line 11: No need to put in a Box the combination of search terms, can be inserted as a phrase within the text. Why did you use plural for ‘models’ and not ‘model*’ or ‘economic crisis*’ as search terms? Possibly could use in search terms also categories of HRH such as doctors, nurses, health professionals, etc. in the search terms too? It is not clear why ‘theoretical’ and ‘models’ were searched, while there are many publications on the effects of economic recessions on health systems including HRH, which might have been informative for this study.

We agree to replace the Box with text-content only, if it fits better the journal’s format or typeset. It is important to take the opportunity to clarify some of the key issues in our search strategy, For example, the term "Models, Theoretical"[Mesh] is an indexed MeSH term, used and applied by trained indexers to each paper present in PubMed. It is not a free-text search term used to detect written content in

titles, abstracts, etc., as we use the tag [MeSH]. Tags determine which type of content the search engine will specifically search for. Unlike free-text terms, indexed search terms, i.e. using the [MeSH] tag, cannot be truncated.

The reviewer refers to the Box 1:

“Box 1: Search Strategy Details in PubMed we used, although unsuccessfully, to try locating any theoretical frameworks specifically addressing the effects of Economic recessions on the HRH.”

That means the search terms in BOX 1 are not the search terms for the main review, but merely a preliminary search to locate any relevant theoretical models for the purpose of generating an priori framework. The full search terms for the main review is rather presented at the Box 2. In that search we do seek for empirical content and we do not restrict for theoretical models. Indeed, we agree with the reviewer we should focus, in our main review, on empirical content.

It is also important to clarify what is implied in use MeSH terms, for example the term "Health Personnel"[Mesh]. As one can observe in the three of MeSH, this single MeSH terms includes multiple MeSH term, for nearly all the health professions (<https://www.ncbi.nlm.nih.gov/mesh/68006282>). Technically, this means that our search included all those MeSH search terms under “Health Personnel”. That is how a PubMed search is run by default, unless specified otherwise.

We understand that search strategies based on MeSH Terms can come across as incomplete to many readers; however, that is visual illusion: a search with the term "Health Personnel"[Mesh] is akin (i.e. provide the same results) that using all several MeSH terms that are part of the broader category of “Health Personnel”[MeSH] - with no need to write them down.

In a simple exercise merely to illustrate the point. In October 7, 2019, we inserted in PubMed the term "Health Personnel"[Mesh] alone: 492354 records came as result. A few second apart, we rather inserted "Health Personnel"[Mesh] OR "Nurses"[Mesh], which means articles focused on one or another, and exactly the same 492354 records were retrieved. One may think the second strategy is more comprehensive than the first but, technically, they are equal. Adding "Nurses"[Mesh] or "Physician"[Mesh] as an alternative to "Health Personnel"[Mesh] is technically unnecessary.

Toward assuring the readers that the search strategy was built by a researcher with proven track record of doing that, we state the following in the main manuscript:

“The search strategy in Box 2 was designed by one of the researchers (TJ) with track record of designing search methods and strategies.36-41. The same author will conduct the searches for the other databases.”

Please also note that we additionally stated the following:

“full search strategy in PubMed, which was additionally checked against the guidelines of the Peer Review of Electronic Search Strategies for systematic reviews.35

21. Page 13, line 50: Again, these two subjects for the review are too vague.

In a few lines below, in the main text, we provided conceptual definitions for each of the main subjects under review – transcribed below:

“For the key terms used, we apply the following conceptual definitions:

- Economic recession: Economic slowdowns affecting one or more countries, where Gross Domestic Product (GDP) contracts for at least two consecutive quarters. Episodes of stagnations, slow growth and less than 2 quarters contractions were excluded.32
- HRH: Any clinical health workers (e.g. physicians, nurses, allied health professionals, pharmacists, oral health or eye health practitioners, community health workers) working in the health, social care or other (e.g. educational, labour) sectors. Clinical researchers are included as they have

a key role for and are often part of the national health systems. Clinical residents, fellows, or students, even undergraduates, are also included, as they are part of the either current or prospected workforce. Such cadres are likely to soon enter into the health labour market, affect its dynamics, and be affected by the impact of economic recessions. Non-clinical staff, such as administrative staff, managers, assistants, other technicians, and health information staff are not necessarily health-specific and excluded from the definition.”

We agree that higher levels of specification are always desirable in the working definitions. It is relatively common, though, that reviews on large, complex topics as this then need to further specify working definitions on issue than were not able to be anticipated at the study protocol stage (e.g. <https://www.ncbi.nlm.nih.gov/pubmed/31553633>; <https://www.ncbi.nlm.nih.gov/pubmed/27436670>). Acknowledging that further specification may be needed in a later stage, we now state the following in the main document, right below the content transcribed above:

“If the definitions above prove to be insufficient for reliable selection decisions, further specification might be added, with consensus among the research team, and then made explicit in the final publication.”

22. Page 14, line 3: It is not clear how else the impact of economic recession on HRH would occur without policies and as a result of the economic recession itself.

We agree the subject needs further clarification. The review aims to shed light precisely on that. Hopefully, we would be able to provide a more competent response to this question after the final publication of the systematic review this protocol reports to.

23. Page 14, line 27: Here HRH is stated that is a key term that includes a variety of professions (e.g. clinical health workers, clinical researchers, etc.), which however were not included as search terms in the review. Why administrative and managerial staff were excluded from the definition of HRH since, apparently, they have an important role in the health system and could potentially have influence patient satisfaction and/or access to healthcare services.

We provided a detailed response above about the inclusion of health personnel in our search strategy. We understand that administrative or managerial staff, or at least a portion of these, can work at the health as well as other sectors of a country’s economy and hence can be affected differently by the health system’s responses to the economic crises. For example, they can move their jobs to other, possibly less-affected sectors of the economy within the same country, while health professionals may need their choices restrained to moving between the private and public health sector, or to external migration for health sector jobs too. Perhaps more importantly, conceptually, we focused on the clinical HRH. That does not imply, though, that we deny the importance of other workforces, working within the health sector, whose specific issues can be addressed by other works.

24. Page 16, line 32: Again, the search terms used for health workers are not exhaustive, could have used more search terms such as specific roles, ‘health workers’, ‘health staff’, ‘doctors’, ‘physicians’, ‘nurses’, etc. There might be a study, for example, on the impact of economic recession on nurses, so this study would not have been included in the findings of this database search with these terms presented in the manuscript.

The answer to this question has been provided above. Studies on “nurses” are included per the search terms, albeit the keyword “nurse” isn’t written down.

25. Page 16, line 34: Why is the search term “Emigration and Immigration” relevant? If relevant, then why the two terms were combined together only and not separately in the search? Not clear why other terms (such as ‘absenteeism’, ‘dual practice’, ‘corrupt’, etc.) were relevant and included in the search terms. There has to be justification backed up with scientific literature for the choice of search terms.

"Emigration and Immigration"[Mesh] is a MeSH term. As explained above, MeSH terms must be spelled out as required by the specific (Pubmed) database. The search terms included reflect, by the

most part, the preliminary model – that is shown in the Figure 2. We acknowledge we were not clear about that before, but we are now (new content underlined):

“The search strategy in Box 2 was designed by one of the researchers (TJ) with track record of designing search methods and strategies,³⁶⁻⁴¹ also supported by the study questions and the preliminary model shown in the figure 2.”

26. Page 17, line 10: It is not clear why authors would search these institutions/organisations’ websites, since these institutions mainly publish technical reports, or else grey literature – which authors stated above that would be excluded from this review.

Point taken. We acknowledge to report the purpose of those searches. We now address the issue in the main text.

“Although we do not include papers from the grey literature, the searches on the institutional websites can help identify relevant empirical, peer-reviewed material for example cited into key reports from the institutional sources.”

27. Page 18, line 19: How is the ‘data extraction table’ different from the framework that would be developed as described previously in the study? Why this ‘data extraction table’ does not include any relevant models or theories potentially presented within the selected articles? When will these be extracted during the review, if not during the screening phase of full texts?

The data extraction table will have two sub-components. One with a pre-planned coding structure for a myriad of (specified) elements of the study characteristics. The other on the data that will inform the refinement of conceptual framework. We have no coding structure (nor a theoretical framework) directly informing the extraction on the latter step. Indeed, we want the data to inductively emerge from the included papers. Only later, will the inductively extracted data be synthesized against the categories of the preliminary theoretical framework. We acknowledge though that we did not specify in the paper all of that we just explained above. Hence, in the data extraction section, we now state (new content underlined):

“The data extraction for the main findings, which consist of either quantitative or qualitative data that could directly inform the refinement of the conceptual model, will be conducted in a different manner. There will be no coding structure, essentially as we want the evidence-based data, arising from the different methodologies included and subject matters addressed, to inductively emerge from the included papers before being analysed and synthesized against the a priori framework. Two independent reviewers (those conducting the Level 2 screening) will extract data, which will be then merged and deduplicated.”

28. Page 20, line 27: It is not clear what is meant by ‘the emergent conceptual model will be tested for sensitivity’ or how this will be performed.

Testing the emergent conceptual model for its sensitivity towards the data is a methodological step within a best-fit framework synthesis approach. We acknowledge we needed to provide further clarification and even examples on the application of this methodological step to this synthesis process in particular. We now do that (new content underlined):

“During the synthesis, the emergent conceptual model will be tested for sensitivity. Indeed, within a “best-fit” framework synthesis, authors need to determine if the synthesis is sensitive to variables such as the reported quality, design or location of included studies.²⁰ For example, we will test (i.e. compare) the effects on the model regarding the use of evidence arising only from high-income countries or from low- and middle-income countries. This may imply adjustments to the model for it to be sensitive or adaptive to varying economic contexts. Similarly, we will the test the model for sensitivity regarding the inclusion or exclusion of the evidence with methodological shortcomings. Along with qualitative saturation principles, such methodological shortcomings will help determine whether those references, findings or new thematic categories will be included or excluded in part or as a whole. However, this will apply only to the papers that are not ‘fatally flawed’.²² Studies with

methodological shortcomings, or areas for which studies do not exist or provided divergent findings also might hint for further research.”

Reviewer: 2

Dear authors,

I appreciate the opportunity to review this protocol, which aims to fill an important gap in the literature by providing a conceptual framework for the impact of economic recessions on human resources for health. I recommend the publication of the protocol and I have just some minor suggestions for the authors:

1. Abstract: I suggest the authors to add "human resources for health" before the respective acronym (line 34);

This has now been corrected. Thanks.

2. Main text: I suggest the author to standardize the terms economic recession/contraction/crisis (e.g. line 82; line 157)

We have now decided to stick to the term “economic recessions” throughout. Occasionally, we maintain the term “crises”, or related words, but not associated to the term economic (e.g. crises’ responses), to avoid repetition of terms in the same sentence, also per the issue noted below.

3. I suggest the authors to re-write the sentence on line 96-97 due to repetition of terms;

The sentence has now been redacted and now reads as follows:

“Scholars have started to explore the area of the effect of Europe’s most recent financial crisis on human resources for health (HRH) policies,¹⁴ and health workers’ responses to the changing economic circumstances,^{9 10 15} but to the best of our knowledge, no specific review has been conducted on this subject.”

4. Please clarify the following paragraph: line 301-304;

This is the same sentence one needed to revise per the comments of the reviewer 1. It now reads as follows (new content underlined):

“The data extraction for the main findings, which consist of either quantitative or qualitative data that could directly inform the refinement of the conceptual model, will be conducted in a different manner. There will be no coding structure, essentially as we want the evidence-based data, arising from the different methodologies included and subject matters addressed, to inductively emerge from the included papers before being analysed and synthesized against the a priori framework. Two independent reviewers (those conducting the Level 2 screening) will extract data, which will be then merged and deduplicated.”

5. If appropriate, a brief reflection on the possible shortcomings/limitations of the best-fit framework synthesis in the context of this study could enrich the manuscript ;

We agree. Early in the Methods section, and after outlining the particular strengths/suitability of that method, we now also state the limitations of the approach (with application to the context of this review) as well as which partial counter-measures have we taken as a result:

“As one limitation, though, the “best-fit” framework synthesis is an emergent method of literature synthesis whose approach is still evolving and being methodologically refined.²⁰ Another limitation is the existence of some subjectivity for the research team in the selection of relevant theories or models for the building of the a priori framework, against which the data will be later synthesized.²⁰ With this respect, and as a partial counter-measure, we will involve a set of experts, not part of the research

team and with varying backgrounds, in the refinements of that a priori framework, before the data synthesis.”

6. Lastly, I would also like to suggest the authors to check the following publication:
Kentikelenis, A.E., 2017. Structural adjustment and health: A conceptual framework and evidence on pathways. *Social Science & Medicine*, 187, pp.296-305.

This conceptual framework refers to the direct effects of austerity measures on the health system, focusing on the healthcare workforce, and may be useful for your work.

We acknowledge the importance of this reference and thank the reviewer for the suggestion. We now include this reference (#8) as part of our Introduction (cited two times) and as part of the theories and models we have considered toward building the a priori framework. Indeed, this paper well synthesizes some of the broader and increasingly known consequences of structural adjustment programs on health issues, taken the case of Greece for illustration. We aim to further advance the knowledge more concretely on the effect on the human resources for health - grounded into a systematic review of the empirical work with such specific focus, and with worldwide coverage.

7. Best of luck with your study.

We would like to thank the reviewer for the insightful comments and encouragement.

VERSION 2 – REVIEW

REVIEWER	Ana Antunes Nova Medical School, Nova University of Lisbon Portugal
REVIEW RETURNED	17-Oct-2019
GENERAL COMMENTS	The authors have addressed all the requests appropriately and I recommend this improved version of the manuscript for publication.